# Southern Horse Mackerel (*Trachurus trachurus*) Spatio-Temporal Distribution Patterns Based on Fine-Scale Resolution Data

**Hugo Mendes \*, Cristina Silva**  **and Manuela Azevedo**

Portuguese Institute for Sea and the Atmosphere (IPMA), Rua Doutor Alfredo Magalhães Ramalho, 1495-165 Algés, Portugal; csilva@ipma.pt (C.S.); mazevedo@ipma.pt (M.A.)
\* Correspondence: hmendes@ipma.pt; Tel.: +351-213027000

**Abstract:** In this study, the distribution patterns of southern horse mackerel are examined using commercial fine-scale resolution data. Using landings by size category and VMS data from the Portuguese commercial bottom-trawl fishery, which consistently targets horse mackerel, this study provides a comprehensive analysis of horse mackerel age distributions spanning a decade (2010–2020). Importantly, this study addresses potential biases in commercial effort data and establishes the usefulness of commercial bottom-trawl gear as a suitable method for sampling and evaluating southern horse mackerel stock dynamics. Ordered regression models were applied to allow for the modelling of the distribution of multiple age categories and investigate spatio-temporal migrations off the Portuguese coast. Southern horse mackerel show a widespread age distribution range and stable abundance with indications of seasonal and spatial patterns in the distribution of specific age groups. The insights derived from this research contribute valuable knowledge for understanding the dynamics and distribution patterns of fish populations.

**Keywords:** commercial fine-scale resolution data; spatio-temporal; distribution patterns; southern horse mackerel

**Key Contribution:** Distribution patterns of fish populations based on commercial fine-scale resolution data.



## 1. Introduction

In fisheries science, understanding and identifying fish habitats is crucial for improving fisheries management. Each marine species exhibits distinct habitat requirements throughout various life history stages, shaped by species-specific traits and ontogeny.

The quantification of patterns of fish distribution can be influenced by the scale at which the observations are made and how data are collected and compiled. Population and community dynamics may show different spatial and temporal structures when the data are observed in different scales. The selection of an appropriate spatial and time scale is crucial for accurately predicting shifts in fish distribution. Processes occurring at smaller temporal or local spatial scales may be unnoticed when relying on data and observations conducted at larger scales. Conversely, processes operating at a larger scale may display gradual variations and be perceived as constant when examined through data and observations at smaller scales [1,2]. In practical terms, the scale of time and space for data is determined by budget constraints. Currently, the understanding of the life history and distribution of many marine organisms in Iberian waters is based on intermittent "snapshots" of species presence, abundance, and distribution, mostly derived from available survey data [3–5].

The horse mackerel, *Trachurus trachurus* (Linnaeus, 1758) is one such fish species that plays an important role in the fisheries and ecosystem dynamics of the Northeast Atlantic. While the distribution and movement of horse mackerel has been investigated in previous studies, a comprehensive study to describe the distribution based on fine-scale temporal and spatial scale data has yet to be completed. The geographical distribution of the horse

mackerel covers the whole platform and slope of the European and African coasts from Norway to the Gulf of Guinea, as well as the Mediterranean and Black Sea [6]. The southern stock population exhibits a geographic distribution spanning the Western Atlantic coastline of the Iberian Peninsula, from the Strait of Gibraltar to Cape Finisterre in Galician waters in the northwestern region of Spain. This stock off the west and southern coast of the Iberian Peninsula has several genetic, phenotypic, and distributional characteristics that distinguish it from the rest of the stocks in the northeast Atlantic [7]. The Portuguese area represents 87% of the total coverage of the stock area and is where the majority of the catches are taken. Moreover, previous studies also indicate that all life stages are present in the Portuguese area supporting the current stock area definition [8,9].

Figure 1 illustrates the Portuguese area, delineating oceanographic zones based on specific geographic and physical attributes. These attributes correspond to distinct oceanographic conditions with the presence of deep canyons creating natural boundaries. These physical features contribute to unique behaviours observed in the biological communities within the designated areas [10,11]. Previous studies [12,13] based on scientific survey data have suggested the presence of diverse distributional patterns within the stock population. A study based on bottom-trawl surveys in winter/spring, summer, and autumn during 1992–1993 [12] analysed juvenile and adult groups differentiated by a 20 cm length threshold. The findings suggested that juveniles mainly distribute in the continental shelf, showing homogeneous distribution throughout the seasons. Conversely, adults have the highest abundance during winter/spring. Another study, based on summer (1989–1990) and autumn surveys (1987–1999) [13], indicated a prolonged spawning season for horse mackerel with the presence of adult spawners along the southwest and south coast during autumn and the continued presence of fish spawning offshore along the coast during summer [13]. A more recent study that extended the analysis of autumn scientific surveys from 1992 to 2015 revealed that during the autumn season, coinciding with the recruitment period, juvenile horse mackerel are most commonly found in the northwestern region with a wider continental shelf [9]. Murta (2008) [8], analysing autumn groundfish survey data, suggested the existence of ontogenic migrations in horse mackerel along the Iberian Atlantic coast, involving two migration paths along the coastline at various depths. Along the Portuguese coast, most year classes initially cluster in the northwest, shifting southward, and occasionally returning to northern waters after reaching seven years of age. The author hypothesized that the migratory movements of horse mackerel were driven by feeding and spawning requirements.

The previous studies were all based on snapshot data from scientific research surveys. However, since the introduction from the European Commission (EC) of legislation [14,15] to monitor European fishing vessels using satellite-based vessel monitoring systems (VMS), the expanding time-series VMS is enabling fisheries scientists to consider the fine-scale spatial and temporal dimensions in commercial fisheries data. VMS allows for the real-time tracking and monitoring of fishing vessels, providing information on their location, speed, and activity. This represents a significant advance in fisheries research.

Vessel monitoring systems are now widely available across Europe for scientific purposes and several studies highlight the potential of VMS for accurately collecting geo-referenced effort data and linking it to logbook and/or observer catch data. However, there are several difficulties associated with the use of commercial data for estimating abundance since commercial fishing vessels tend to target specific areas [16,17], as well as other challenges such as overestimation and synchronization that still need to be addressed [18–20]. Data from the VMS can offer a comprehensive set of indicators to improve inputs for stock assessments, enable real-time distinction of fishing grounds, and facilitate the assessment of regulatory measures. It could also be used to evaluate the effectiveness of marine protected areas and to inform the design of spatial management measures [21] and to support the development of ecosystem-based fisheries management [22,23].

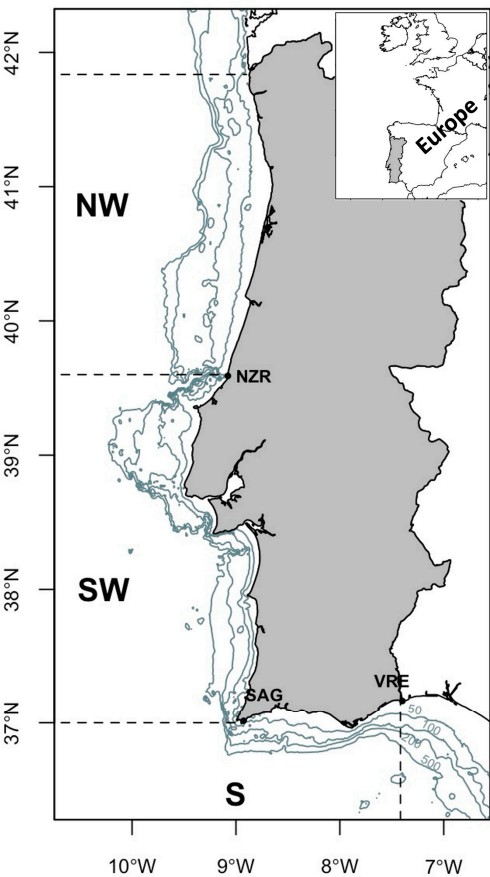

**Figure 1.** Study area. Portuguese Iberian Peninsula, showing depth isobaths (50, 100, 200, and 500 m) and the three distinct geographic areas defined by noticeable geomorphological features.

Azevedo and Silva (2020) [24] combined VMS data with species sales notes by commercial size category and biological information from onshore sampling to investigate the potential of this fine-scale spatio-temporal resolution in biological, fishery, and effort data to assess horse mackerel distribution patterns by life-stage. The current study expands the temporal scope of this work to analyse age specific patterns of seasonal and inter-area migrations and to assess both intra- and inter-annual fluctuations in the species distribution. Horse mackerel is the main target species of the Portuguese bottom-trawl fishery providing a valuable source of data for understanding the dynamics of the southern horse mackerel population. The fine-scale resolution data obtained from this study offers insights into the spatial and temporal distribution patterns of the species, as well as the fishing effort exerted to capture them. In addition to the fishing effort, the catch-at-age composition of southern horse mackerel provides valuable insights into the population structure and dynamics.

By understanding the dynamics and distribution patterns of fish populations, fisheries managers can make informed decisions about sustainable fishing practices and conservation efforts. The framework used here has the potential to provide valuable background for the management and conservation of commercially exploited fish stocks in the Northeast Atlantic.

## 2. Materials and Methods

### 2.1. Data Exploration

From 2010–2020 bottom trawling consistently constituted the dominant gear for harvesting this species. The landings of horse mackerel in Portuguese auctions are performed by size categories, following EU regulation [25]. This regulation establishes the common marketing criteria for fishery products in the EU. Within the framework of this regulation, at auctions, each box is assigned a fish size category ranging from T1 to T7. Boxes labelled

as T1 include the largest fish and those labelled as T7 contain the smallest fish. Horse mackerel landings distribution by size category from 2010–2020 is primarily among the size categories ranging from T3 to T5. However, in certain years we observed substantial landings of the smaller size category T6 and the larger size category T2. To describe the spatial and seasonal patterns of this fishery, horse mackerel catch length composition along the Portuguese coast was estimated by fishing trip using the estimated length composition by size category obtained from the onshore sampling scheme and following the raising methodology detailed in Azevedo et al. (2020, 2021) [24,26]. In summary, horse mackerel onshore sampling during 2010–2020 was carried out along the year in 16 landing ports spread along the Portuguese coast. The length composition (1 cm size classes) by size category in each year and port visit was estimated by raising the length distribution from the sampled size category to the size category landings in that year and port visit. The mean length composition by size category and area (NW-SW-S) was obtained by pooling, for all sampled ports in each year and area, the raised length composition. The mean length of horse mackerel across different size categories exhibited notable stability throughout the period from 2010 to 2020. This consistency is indicative of economic reasons, as maintaining stability in length aligns with economic interests, especially given the significant variations in the price of horse mackerel by size categories. A detailed description of the landings distribution by fishing gear, size category, and mean length of horse mackerel across different size categories from 2010–2020 can be found in the Supplementary Materials (Figures S1–S4).

Discards are estimated by Portugal since 2014 from national at-sea sampling programs on board commercial vessels operating in the area. Discards are usually very low and infrequent so as to be considered negligible [27]. For the purposes of this study, we will use the terms landings and catch interchangeably in reference to the data.

Horse mackerel bottom-trawl landings and size category were coupled with the corresponding bottom-trawl VMS trip records to characterize the spatial distribution of the landed size categories along the Portuguese coast (Table 1). Bottom-trawl effort was estimated by applying vessel speed criteria between 2 and 5 knots [28] to identify VMS records that correspond to fishing activity. Catch data by trip are assigned to the trip VMS positions based on the sales date recorded at auction and the assigned landing date. The total landings by size category assigned to each trip were uniformly distributed by all VMS fishing records of the trip, proportionally to the effort in each point. The assignment of horse mackerel size categories differed among areas. Therefore, to correctly allocate bottom-trawl trip landings to each area, only single-area trips were considered, accounting for 77% to 87% of horse mackerel bottom-trawl catches from 2010 to 2020. Depth information of fishing activity locations was added to the VMS records by overlaying them on a 1 min depth grid layer obtained from the website Satellite Geodesy (Global Topography, Measured and Estimated Seafloor Topography, https://topex.ucsd.edu/cgibin/get_data.cgi (accessed on 15 November 2021)). Duplicate records and spurious values in position, speed, effort, and catch were identified along the process and removed from the subsequent analysis. The catch-at-length composition by trip was subsequently converted into catch-at-age, based on spatially stratified age sampling data (NW-SW-S) collected within the EU Data Collection Framework [29], in which individual fish ages are determined by counting the growth bands in otoliths following the guidelines from the ICES age reading workshops [30]. Since, according to the birth date convention, the youngest fish caught in the second semester are aged "zero" [31,32], specific semester Age-Length Keys (ALKs) were computed by year in the period 2010–2020 and applied to the estimated catch-at-length in the corresponding semester. Fish of 11 years and older were pooled into age 11+ as used for stock assessment purposes [26]. By converting the catch-at-length composition into catch-at-age data, this study aims to unravel the age composition patterns of the species. The analysis of age groups and their seasonal and depth patterns can offer a deeper understanding of the life history and behaviour of southern horse mackerel, shedding light on their spawning seasons, migration patterns, and habitat preferences.

**Table 1.** Description of the aggregated data used for analysis in the period 2010–2020.

| Data | Time Aggregation | Spatial Aggregation | Basis |
|---|---|---|---|
| Catch-at-length (number) | Daily | 0.05° × 0.05° | Landings (weight) by trip recorded at auction by size category × length distribution by size category |
| Catch-at-age (number) | | | Catch-at-length by trip × semester Age-Length Keys by year |
| Effort (kW × hour) | | | VMS data (trawl hours by fishing position) × vessel power information (from EU Fleet Register) |
| Depth (meters) | | | Satellite Global Topography |

Table 2 presents the final set of data used in this study with key information about the fishing activity from 2010 to 2020 including the number of boats, total trips, trawl hours, average engine power (measured in kilowatts, kW), and average depth (meters). Analysing this data provides insights into how fishing operations have evolved during this time. The number of active vessels has remained relatively stable, fluctuating between 34 and 43. This stability is reflected in both the total number of trips and trawl hours. While the average engine power shows a slight decline over the analysed period, the average trawling depth remains close to 100 m. The absence of noticeable trends or significant changes in these factors suggests consistent fishing intensity and strategies. The lack of substantial shifts in the data has important implications for interpreting catch data, indicating a certain stability in fishing practices over the studied period.

**Table 2.** Summary of data (vessels, effort, depth, and catch) in the period 2010–2020.

| Year | Number of Vessels | Total Number of Trips | Total Trawl Hours | Average Engine Power (kW) | Average Depth (m) |
|---|---|---|---|---|---|
| 2010 | 38 | 4352 | 53,683.1 | 538.9 | 97.6 |
| 2011 | 35 | 3842 | 52,479.5 | 529.1 | 99 |
| 2012 | 37 | 4412 | 54,449.0 | 529.3 | 107.5 |
| 2013 | 34 | 4093 | 46,158.2 | 523.8 | 114.1 |
| 2014 | 36 | 4279 | 53,240.3 | 526.8 | 119 |
| 2015 | 43 | 4537 | 59,361.8 | 519.6 | 111.8 |
| 2016 | 43 | 4778 | 52,682.5 | 516.7 | 114.4 |
| 2017 | 39 | 4577 | 61,092.6 | 515.3 | 110.9 |
| 2018 | 42 | 4511 | 64,709.4 | 514.1 | 102.3 |
| 2019 | 43 | 4721 | 65,473.4 | 495.5 | 94.9 |
| 2020 | 42 | 4736 | 68,020.0 | 465.1 | 100.9 |

*2.2. Effort Data Distribution*

Indices of relative abundance and composition data representing the proportions of the sampled population within different age, length, sex, or weight categories directly inform trends in population biomass [33]. Catch-per-unit effort (CPUE) is a measure of the amount of fish caught per unit of fishing effort, such as the number of fish caught per hour of fishing, and is often used as an index of abundance in stock assessment models. Fishery-dependent indices are subject to various factors that can challenge the assumption of proportionality to abundance [34]. One of the major challenges when utilizing these indices is that fishing effort is not uniformly distributed in time and across the stock area. Instead, it tends to concentrate in regions where fish abundance is higher and when market demand is higher. To address these issues, exploratory analysis was performed on the spatial distribution of daily fishing effort measured in trawling engine power × hours (kW×hour). The exploratory analysis also aimed to assess potential changes in fishing effort and targeting strategies that could affect the fishery catchability from 2010 to 2020.

A detailed description on fishing effort distribution and selectivity can be found in the Supplementary Materials (Figures S5–S8).

Results show that fishing effort was stable over the analysed period, with no significant trend observed (Figure S5). Catches of horse mackerel have remained relatively stable over time. Despite the increasing fishing opportunities for this stock, the fishing industry has maintained a consistent effort level towards horse mackerel in order to stabilize its commercial value [35]. Commercial fishing vessels often concentrate their efforts in specific target areas where fish are abundant. Target fishing grounds for trawlers are also conditioned by morphological traits and protection legislation, such as operation restrictions in areas near the shoreline, which can lead to overestimation of abundance in some areas and underestimation in other areas and/or ages. Although age 0 individuals do not seem to be well recruited in the trawl fishery; overall, the total catch by age of the horse mackerel trawl fishery has not changed significantly from 2010–2020 with a consistent pattern of catches and most occurring between ages 1 and older (Figure S7). This could indicate a lack of significant changes in factors affecting fishery catchability, such as changes in the fishing gear and targeting strategies. This is important as a varying selectivity for different sizes of fish can introduce bias in our estimates of abundance, whether measured by length or age [33].

Figure 2 shows the distribution of effort by weekday with a notable increase in effort during the middle of the week followed by a subsequent decrease. The auto-correlation function (ACF) measures the correlation between the amount of effort and its lagged values. Fishing effort exhibits a strong autocorrelation in consecutive days, indicating a persistent relationship with its past values. The combined effort distribution and ACF reveal a consistent weekly cycle, indicating a recurring pattern every seven days throughout the analysed period. Together, these analyses provide insights into the temporal dependencies within the fishing effort data. This persistent weekly cycle in fishing effort can be attributed to the economic weekly cycles of buyer's demand in the auction market wherein commercial fishers typically align their effort. Commercial fishers often strategically align their fishing efforts with these weekly demand fluctuations, resulting in the pattern observed in the autocorrelation function.

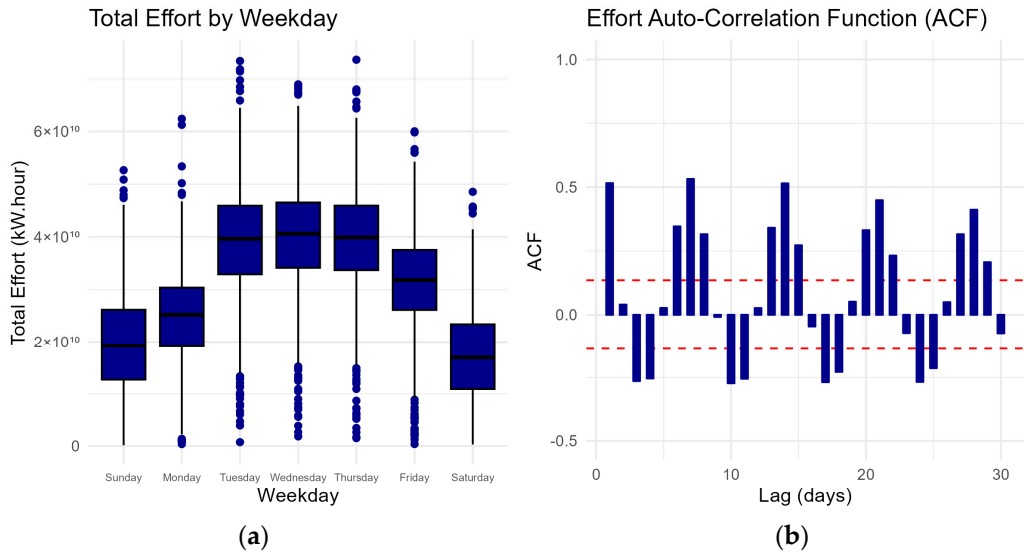

(a) (b)

**Figure 2.** (**a**) Boxplot of the distribution of effort by weekday and (**b**) auto-correlation function between fishing effort at different daily lags. Fishing effort is correlated each consecutive day and shows a strong weekly cycle. Dashed red lines indicate the threshold for statistically significant autocorrelations.

Spatial distribution also showed to some degree a cyclic behaviour evident in the distribution of effort by latitude depicted in Figure 3. Although a large coverage of the

spatial trawl effort is available, there are patterns in the total amount of effort exerted across the identified fishing grounds. For example, the fishing grounds among the defined geographic areas (NW, SW, and S) are clearly divided by geomorphological traits where no effort is exerted. The number of vessels identified and consequently the effort in the southern region was clearly inferior to the other areas (Figure 3). Moreover, horse mackerel fishery distribution in the southern region is characterized by the widespread presence of adult fish ranging from coastal waters to over the slope [24]. We opted to exclude the southern region from subsequent analyses because of the low effort and catches, and we identified specific and narrow fishing grounds that could lead to inadequate coverage of this area. This exclusion was considered necessary to mitigate the potential introduction of significant bias to the age distribution within the study area and our analysis concentrated on the two NW and SW areas with a broader coverage of fishing effort.

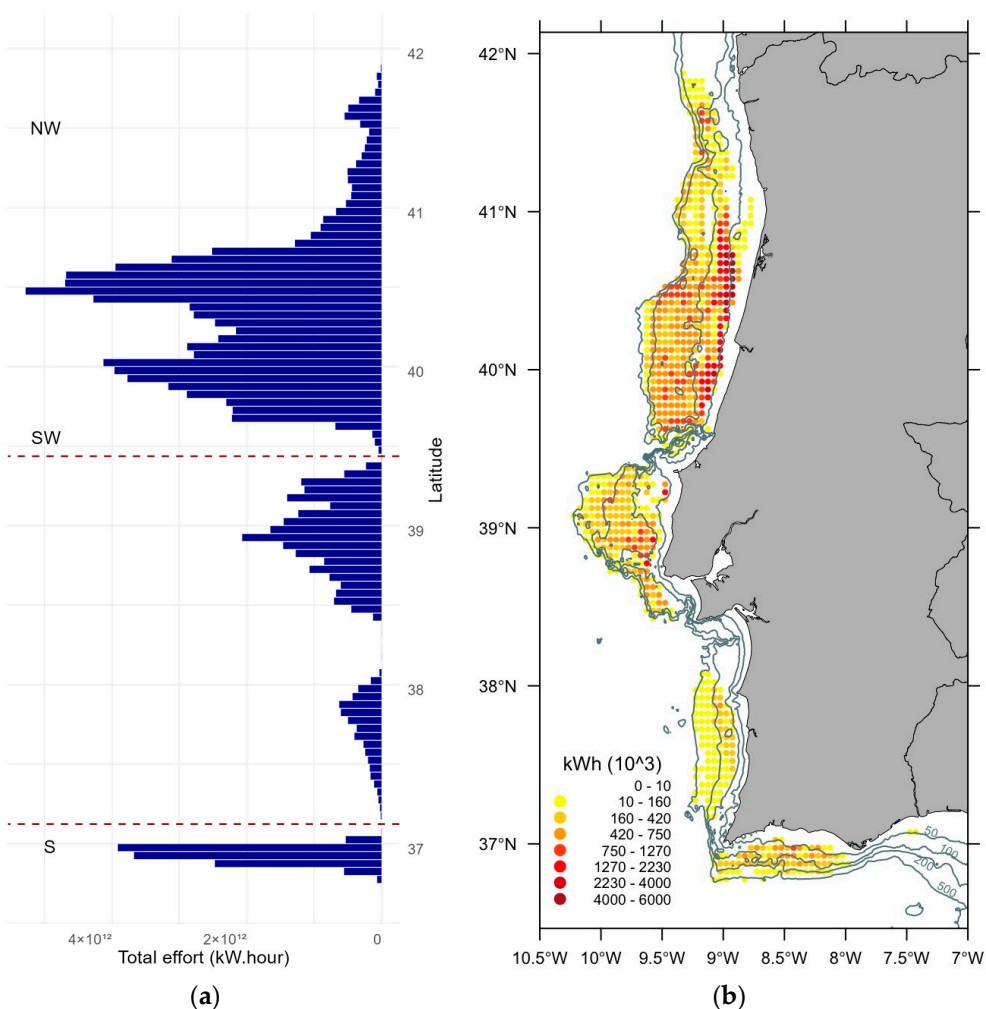

**Figure 3.** (**a**) Distribution of effort (kW × hours) by latitude in NW, SW and S areas and (**b**) corresponding spatial distribution in 2010–2020 showing the 50 m, 100 m, 200 m, and 500 m isobaths.

There were several limitations associated with using commercial effort data for estimating abundance. Time and spatial non-randomness were observed in our fishing effort data, characterized by a pronounced weekly cycle and under-sampled and/or unsampled grid cells. Given the robust weekly pattern in fishing effort associated with market week cycles, effort values were averaged on Julian weeks and for each geographic area. Our simple average smoothing approach is expected to overcome the issues referred to above and we considered it appropriate to standardize our effort data.

Integrating the VMS effort data with the onshore sampling scheme for horse mackerel, coupled with biological and commercial data, yielded spatial weekly catch-at-age abundance indices encompassing age groups from 0 to 11+. Subsequent analysis of this dataset aims to evaluate distributional and migratory patterns, as previously indicated in studies, and support the effectiveness of this framework in optimizing the use of high-resolution spatio-temporal commercial data. In addition to having an age abundance index, understanding the spatial boundaries of exploited stocks is crucial for any stock assessment. The subsequent analysis of distributional and migratory patterns aims to contribute to the development of improved methods for defining stock units and promoting evidence-based management.

### 2.3. Statistical Analysis

The age composition catch data from ages 0 to 11+ can be viewed as an ordered categorical response, comprising the count of individuals in each age group normalized by fishing effort (in kW $\times$ hours) to achieve a CPUE abundance index. If there are only two groups present (e.g., juvenile, adult), the response can be considered binary. In this scenario, standard logit transformation and modelling tools such as generalized linear models (GLM) or generalized additive models (GAM) can be applied, as previously tested by Azevedo and Silva, 2020 [24]. Conversely, in cases where more than two groups are present, the response is multinomial, and a standard logit transformation is unsuitable. In our study, the variation in multinomial catch-at-age data was performed by continuation-ratio logits. In the multinomial case, a response probability is described by multiple logits and a unique feature of continuation-ratio logits is that the various logits for a response can be viewed as logits for independent binomially distributed data [36]. Continuation-ratio logit models (CR models) have an advantage over other ordinal regression methods in that it is very easy to remove or loosen the proportional odds assumption. This allows for all covariates or some subset of covariates to be able to freely vary with every level of a category [37]. This is particularly useful since the proportional odds assumption is not accomplished for every age group derived from the trawl nets selectivity.

The abundance at each age group, $a = R \ldots A$, where $R$ denotes the recruitment age and $A$ denotes the oldest age class category, was modelled as the conditional probability of being of age $a$ given that it is at least age $a$, $P(Y = a | Y \geq a)$. The CR model is then well suited to model the distribution of age abundance through $A$ minus $R$ models and the continuation-ratio logits are defined as [37]:

$$log\left(\frac{P_a}{P_a + \ldots + P_A}\right), \ a = R \ldots A - 1 \tag{1}$$

Continuation-ratio logits have the particular feature that the different logits for a response can be regarded as logits for independent binomially distributed data. Each logit can then be analysed separately by means of a generalized linear model. This approach enables the application of generalized linear models separately to each level of the logits, facilitating the analysis of the variation in multiple age categories [38].

We combined CR models with generalized additive models, allowing for the analysis of ordered multinomial responses with separate linear or non-linear terms for each age category using Vector Generalized Additive Model family functions [39,40] for fitting VGAM models. A VGAM model was fitted using a logit link as a function of the discrete variable fishing area (Northwest and Southwest), and in this study we also describe how smooth functions of continuous variables depth and Julian week are suitable for describing each age distribution CR model.

Individual linear and non-linear effects on each combination of the response and predictor variables were performed to analyse the parallel assumption (i.e., predictor variables having a consistent effect across the different levels of the response variable). If, for example, the probability of age abundance at different age levels is assumed to change at a constant rate (or slope) with variations in our predictor variable, then the parallel

assumption is met. In instances where the parallel assumption was violated, a modification of the CR model was applied where a set of predictor coefficients was estimated for each individual age category. This flexibility is particularly useful when the parallel assumption is not met by accounting for potential differences in the predictor variable effects across different categories of the response variable, but it can be computationally challenging [40]. Several combinations of predictor variables were tested on parallel, non-parallel, or partial parallel models and the selected age distribution model was chosen based on the deviance explained and the AIC values.

All statistical analyses were carried out using the R software, version 4.3.2 [41], as well as the extension package VGAM, version 1.1.8 [40]. The spatial data was analysed with the package sp, version 1.6.0 [42] and the visualization was performed with ggplot2, version 3.4.2 [43].

## 3. Results

### 3.1. CPUE-at-Age Distribution

The age profile shape of the trawl commercial catches remained consistent from 2010–2020, with no evidence of changes in the trawling gear used during the analysed period (see Figure S8 in the Supplementary Materials). The majority of catches occurred between ages 1 to 6, as these ages appear to be better represented in the catch-at-age data (Figure 4). There could be several factors contributing to this pattern, including the greater availability of these age groups for the specific fishing grounds and the gear used by the horse mackerel trawl fishing fleet. Still, the decreasing trend observed with ageing fish is to be expected by the increased mortality and lesser availability to the fishery and provides some confidence that our abundance indicator is able to follow horse mackerel cohorts. Age-0 individuals do not seem to be well recruited in the trawl fishery, with a lower mean CPUE when compared to age-1, suggesting a potential gap in the abundance index for this specific age category.

Visual inspection on the mean catch per unit effort at different ages (CPUE-at-age) between 2010 and 2020 shows inter annual temporal variability (Figure 4a). The CPUE-at-age exhibits fluctuations across these years, with some values clearly above the average in successive years. This could suggest the presence of robust year classes, where certain age groups of the fish population exhibit notably higher catch rates over consecutive ages. Another key aspect revealed by the CPUE analysis is the existence of age specific seasonal patterns as shown by our aggregated data by Julian week. Variability in CPUE-at-age is evident over different weeks, suggesting age dependent seasonal variation (Figure 4b). The combined average catch per unit effort at different ages reveals distinct seasonal patterns for various age groups. Younger fish at age-0 and age-1 appear to be more available during the winter and autumn seasons. For age-2 and age-3, characterized by a mixed composition of mature and immature individuals, a less defined seasonal pattern is observed, with their occurrence primarily from the beginning of the year until early summer. In contrast, fish at age-4 and older exhibit increased mean abundance in the spring season, suggesting a potential association with their increased availability to fisheries during the spawning season.

The visual inspection of aggregated CPUE-at-age by geographic area already reveals some patterns. Notably, the NW area is characterized by a concentration of younger individuals, while a gradual shift in mean CPUE of older individuals is observed towards the SW area (Figure 4c). The CPUE-at-age over the three different depth strata provides some insights in the age distribution of horse mackerel individuals by depth, revealing the stratified nature of the population resulting from ontogenic migrations. Younger individuals predominantly occupy shallower strata, gradually shifting to deeper strata with age. However, even though older individuals (beyond age 5–6) prefer deeper waters, there are also occurrences of younger individuals in these deeper strata (Figure 4d).

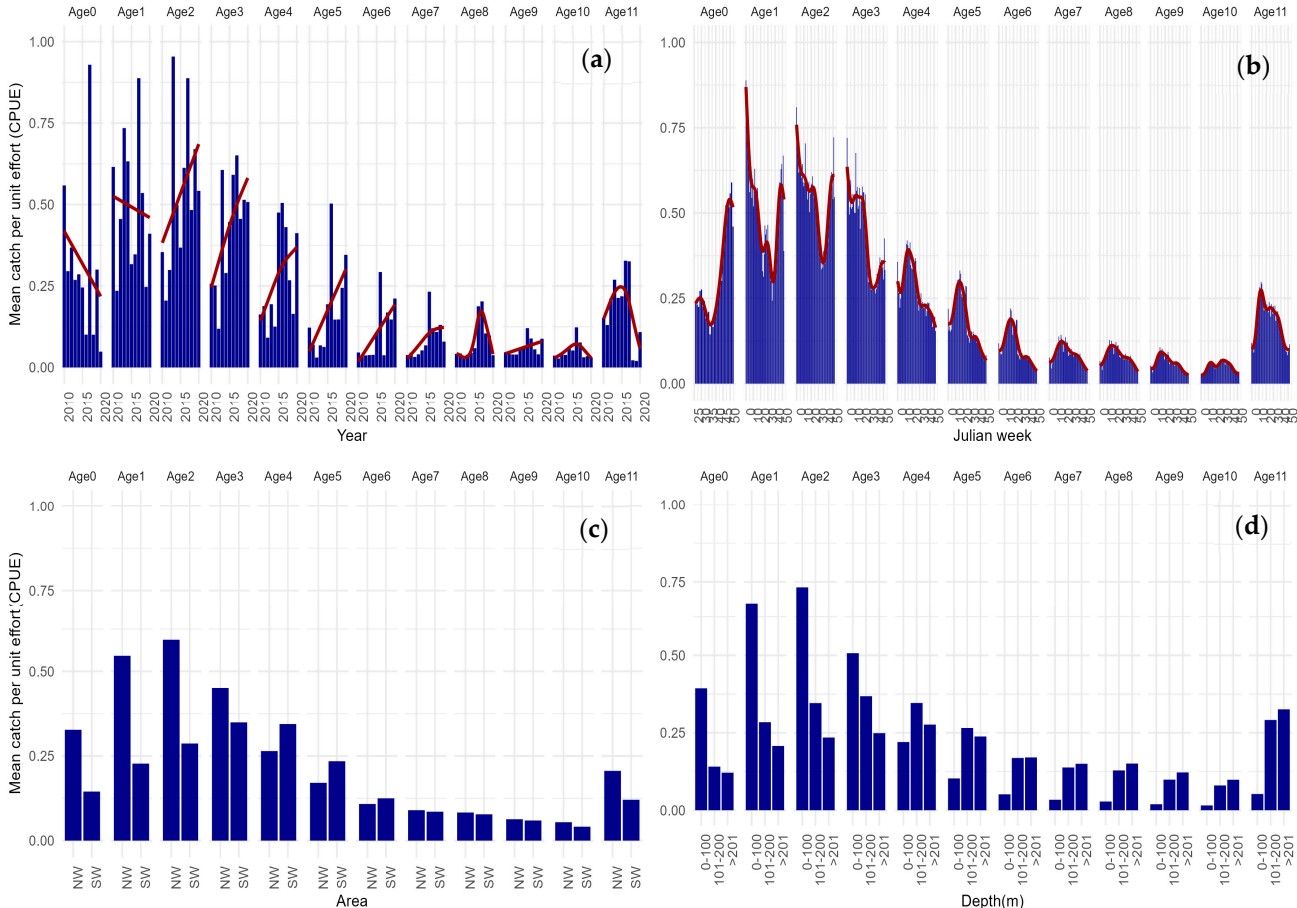

**Figure 4.** Mean catch per unit effort from age-0 to age-11+; (**a**) age variability between 2010–2020; (**b**) age seasonal variability by Julian week; (**c**) regional variability in NW and SW areas; (**d**) age vertical distribution by depth strata, 0–100 m, 101–200 m, and >201 m. Smoothed trends are represented by the red lines.

To further enhance our visual exploration of age-dependent migrations, Figure 5 illustrates the proportion of CPUE-at-age across Julian weeks for each year, categorized by depth levels, 0–100 m, 101–200 m, and >201 m and in the NW and SW geographic areas. By examining the diagonal panels in each age/year plot, we can effectively trace the proportion of each age/year class within the spatial variables of depth and area throughout the Julian weeks of each year and provide a more detailed illustration of the age specific distribution, without relying on the filtered aggregated mean abundances. This approach shows a more detailed depiction of how age distributions vary across both temporal and spatial dimensions. The plus group age 11+ was removed from the analysis because this group consists of surviving members from previous ages resulting in a combination of several cohorts in this group that could confound the analysis. Figure 5 also highlights the trajectory of the 2010 cohort, providing a representation of its path over time. Additionally, in Figure 6, we present the spatial distribution of the 2010 cohort, offering an illustration of this cohort's geographical dispersion as an example.

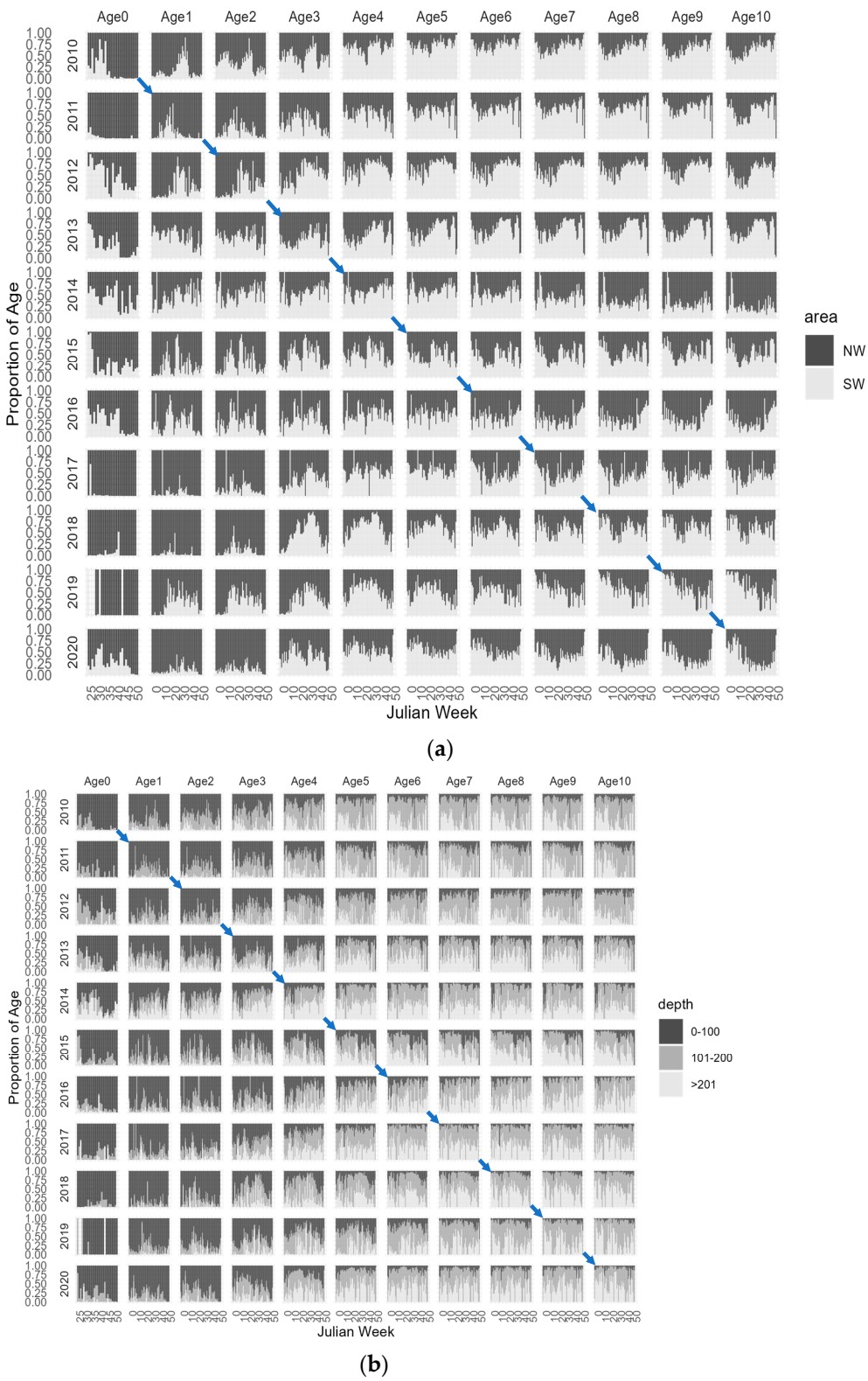

**Figure 5.** Weekly CPUE-at-age from 2010–2020 in (**a**) Northwest (NW) and Southwest (SW) Portuguese areas and by (**b**) depth levels, 0–100 m, 101–200 m, and >201 m. The diagonal panels in each age/year plot trace the proportion of each age for a given year class. Blue arrows represent the 2010 cohort trajectory from age-0 to age-10.

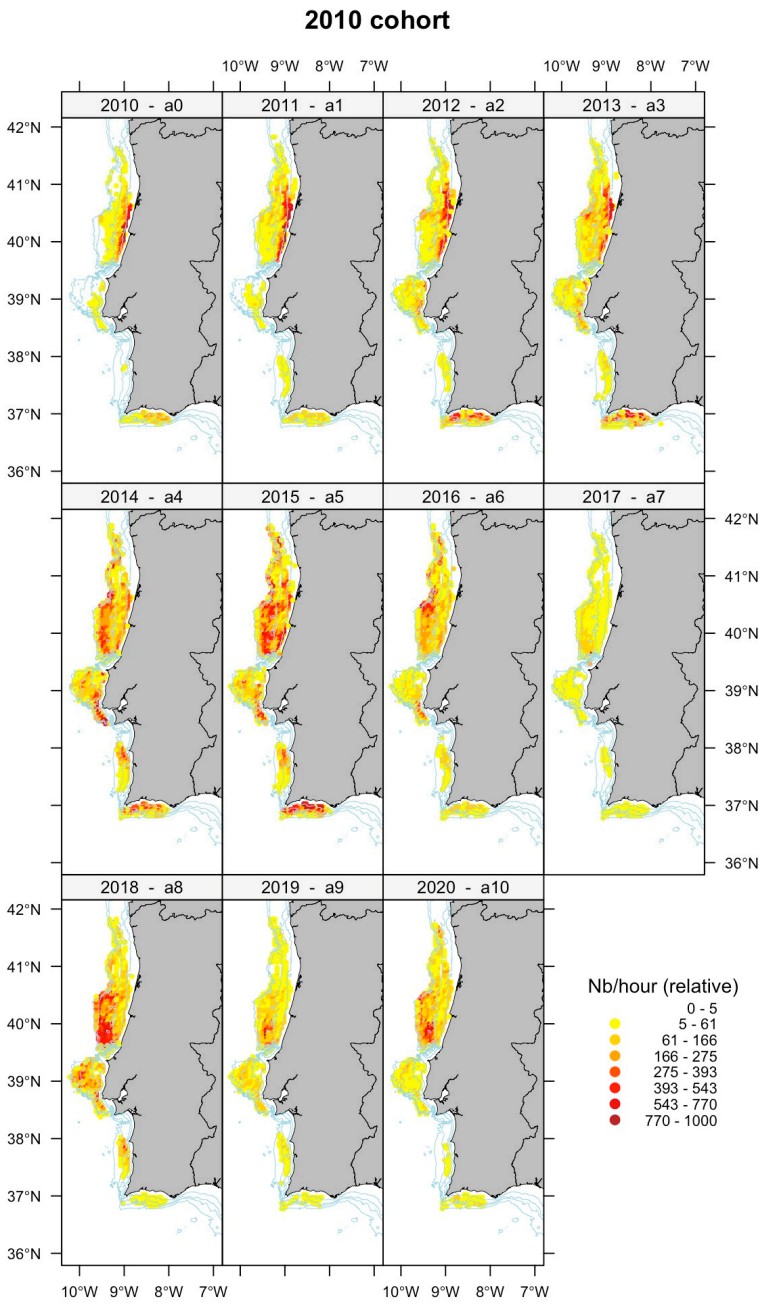

**Figure 6.** Abundance distribution from age-0 (**top left**) to age-10 (**bottom right**) for the 2010 cohort. CPUE is standardized in each panel relative to the maximum number of each age group.

Examining the distribution of age groups based on the proportion of abundance in the two studied geographic areas reveals some distinct patterns. Although not fully recruited in the trawl fishery and only available from the second semester, it is evident that the majority of age-0 individuals are concentrated in the northwest area. Age-1 and age-2 also exhibit higher availability in the NW region, with some concentrations observed in the SW area. Age-3 appears to have an equal distribution in both regions, showing a more pronounced seasonal pattern in the SW area, where the majority of this age class is prevalent in the mid-year. Older individuals demonstrate a higher prevalence in the SW region, although in specific age/year classes, and they are also observed in the NW area. Analysing the trajectory of each year class enables our hypothesis of younger ages in the northwest area that progressively move to south before returning to north at older ages.

Moreover, there appear to be consistent trends and patterns among the various age/year classes that suggest the presence of stable migratory behaviours.

Analysing the proportion of age abundances in three depth strata, 0–100 m, 101–200 m, and >201 m, shows that younger age groups, from age-0 to age-2, are primarily concentrated in the shallower strata, although there are also observable occurrences in the deeper strata. Age-3, on the other hand, displays a more balanced distribution across all three strata, with some variability observed among different year classes. Age-4 and older individuals are much more frequent in the intermediate and deeper strata with minimal occurrences in the shallower strata throughout the year. The prevalence of older ages at deeper strata provides evidence for the hypothesis of ontogenic migration to deeper areas.

The visual representation of both aggregated and proportional age abundance indices has revealed consistent trends and patterns within different age/year classes in our data. These observations indicate the existence of stable migratory behaviours along the western coast, extending to various depths.

### 3.2. Model Results

Using the CPUE-at-age as an abundance indicator for horse mackerel, we explored the age probability distribution of horse mackerel within the available cohorts in our data. Age-0 was removed from the CR models because this age class is not completely recruited to the trawl fleet. The trawl gear used in this study does not effectively capture the abundance of these younger individuals. This decision was made to avoid biases in the models and concentrate on the older age classes available to the trawl fleet, which are more likely to reveal distributional and migratory patterns.

The overall similar patterns of age proportions (Figure 5) has revealed consistent trends and patterns within different age/year classes in our data. Exploratory modelling was performed on individual cohorts with both above-average (2011, 2012) and below-average (2010, 2013) recruitments, as indicated in the ICES assessment [26] that also suggested that year-class strength does not influence significantly the age distribution considered in this study during the analysed period.

Parallelism or proportional assumptions were checked before proceeding with the CR model. Preliminary exploration suggested an intermediary model between the parallel and the non-parallel model where only some of the explanatory variables are parallel and others not. Detailed outputs of the VGAM function for the CR models tested, including parameter estimates, significance, and model diagnostics, are available in the Supplementary Materials (Tables S1 and S2). From the several models tested, the selected model can be described as follows:

$$logit\ P(Y = a | Y \geq a) = \alpha_a + \beta_a \text{area} + s_a(\text{Julianweek}) + s(\text{depth}) \tag{2}$$

where $s()$ represents vector cubic smoothing splines for each age and the conditional probabilities are modelled as a function of the categorical variable area and smooth terms for continuous variables Julian week and depth, treated as unparallel and parallel processes, respectively, in the model. It states that the area and Julian week variable effects should not be the same in all the categories, as opposed to depth where the effect is the same across all the age categories. Because the parallel regression assumption is not met for area and Julian week, the resulting model incorporates estimated parameters specific to each level of the response variable (refer to Table S2 in the Supplementary Materials).

The CR model predictions revealed that age probabilities across geographical areas have distinct patterns. In the younger age groups (age-1 and age-2), there is a higher conditional (relative to other age groups) probability of occurring in the NW by 0.18 and 0.21, respectively, compared to 0.10 and 0.14 observed in the SW area. Age-3 shows a similar median probability of occurring between areas (0.18) with overlapping ranges. From age-4 to age-6 a shift in area occurs, with these age groups becoming more prevalent in the SW, resulting in conditional probabilities ranging from 0.18 to 0.08. Older age groups exhibit comparable probabilities in both areas with overlapping ranges (Figure 7a). These results

align with the analysis of the CPUE-at-age by area shown in Figure 4c, suggesting that our combined age probabilities model is capturing the overall pattern observed in the mean CPUE during the period 2010–2020 and may imply that there is a consistent pattern of migrations during the analysed period. A reliable indicator of age-dependent migrations is the increase or absence of decrease in the abundance of a specific age/year class within a given area, accompanied by a corresponding decrease in the abundance of the same year class in an adjacent area.

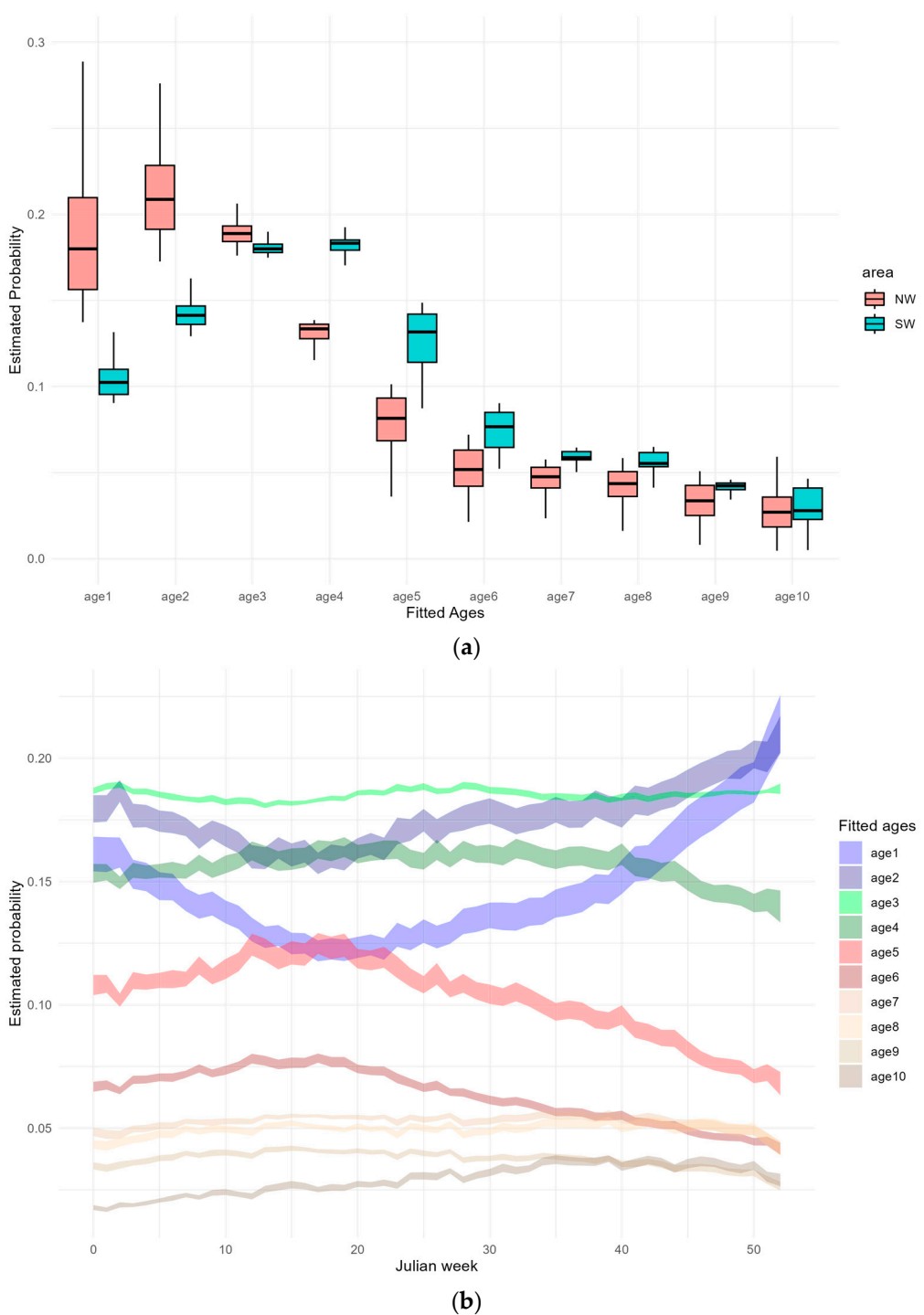

**Figure 7.** *Cont.*

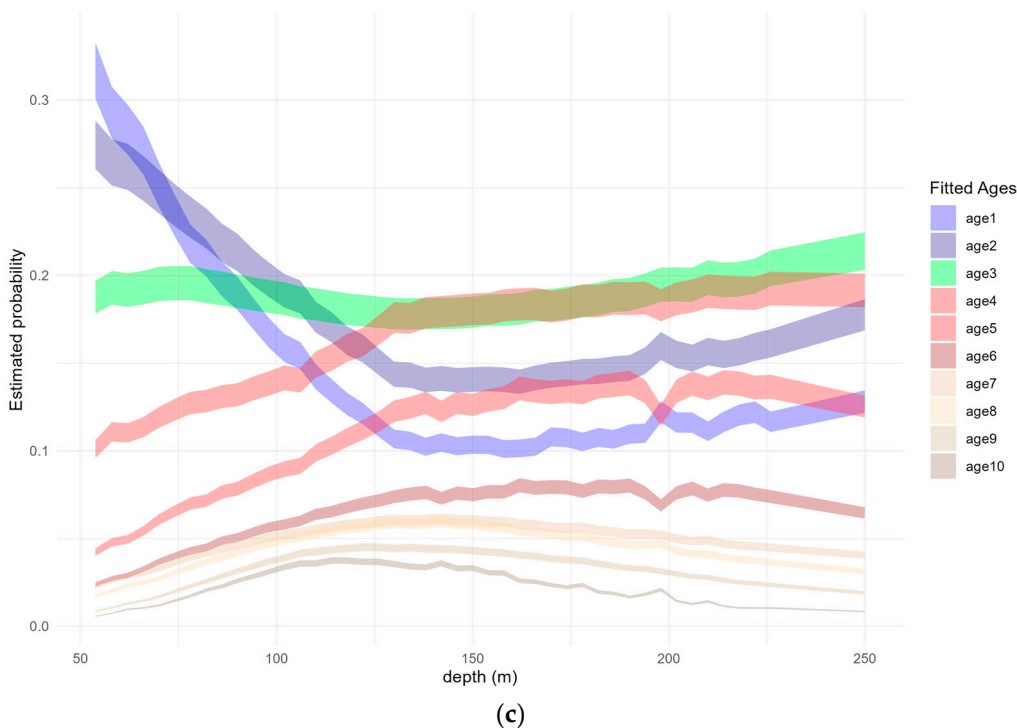

**(c)**

**Figure 7.** (**a**) Boxplots of predicted age probability of occurrence in each of the categorical area levels and average predicted probability with 95% confidence interval by (**b**) Julian week and (**c**) depth. (Outliers not shown in the boxplot).

Furthermore, the examination of age distribution across Julian weeks exposes a few seasonality patterns. Age-1 and age-2 share similar patterns, showing higher probabilities of occurrence during autumn and winter, peaking at maximum probabilities of 0.21 each. Age-3, with an average conditional probability of 0.18, and age-4, averaging 0.16, display a less pronounced seasonal pattern, with relatively uniform probabilities throughout the year. Age-5 and age-6 increase in the probability of occurrence from the beginning of the year until early summer, reaching maximum probabilities at 0.12 and 0.08, respectively, and decreasing until early winter with a minimum of 0.07 and 0.04. Older individuals exhibit constant probabilities all year (Figure 7b). The analysis of the mean CPUE-at-age from 2010 to 2020 by Julian week (Figure 4b) shows some similarities with our model results. However, the patterns identified in the mean CPUE were more pronounced and exhibited similarities between age-0 to age-3, characterized by a decline in abundance during spring/summer and an increase in autumn/winter. Another single pattern emerged for age-4 and older, with an increase in abundance during spring/summer.

Age distribution with respect to depth also reveals distinct patterns across different age groups. Age-1 and age-2 exhibit a similar pattern, characterized by higher abundance in shallow waters, gradually decreasing the probability with depth, reaching 0.097 and 0.136 at 150 m, respectively, although stable occurrences are also noted below this depth. Age-3, on the other hand, displays a relatively consistent estimated probability across all depth levels, averaging 0.19 all year. Age-4 to age-6 share a common pattern, indicating reduced presence in shallow waters and a gradual increase in occurrence with depth, peaking around 125 m and maximum probabilities 0.19, 0.13, and 0.08, respectively. Older age groups follow a similar pattern with increasing occurrences until the depth range of 125 to 150 m and a slight decline in abundance at greater depths (Figure 7c). The age probabilities predicted by the model align with the distribution of mean CPUE-at-age from 2010–2020 across the three depth strata (Figure 4b). Although the selected CR model was fitted assuming parallelism in the depth effect, a deviation from this effect is evident in the

predicted estimated age probabilities. This distinct pattern, observed in the data, may be attributed to the combination of the several predictor variables included in the model.

## 4. Discussion

This study on the age distribution patterns of horse mackerel reveals important insights into the challenges associated with commercial fishery data collection. This study employs a fine-scale framework that exposes seasonal variations and patterns in the abundance of specific age groups, contributing to a comprehensive understanding of the spatial and temporal dynamics of age distributions in fish populations.

The continuation-ratio logit model proved to be a robust analytical tool for investigating successive age distributions. As a variant of an ordered regression model, this approach is particularly well-suited for modelling processes that unfold in stages, especially those in which it is not possible to return to a previous stage [44]. This model provides a practical framework for understanding sequential processes where the response takes on successive stages over time [40]. Unlike other well recognized ordinal models such as cumulative models, these models do not encounter structural issues related to out-of-order cumulative probabilities or substantive dispersion effects [37]. Therefore, they could be applied effectively to our commercial trawl data, which is anticipated to contain some under-sampling of certain regions and depths and consequently age categories. The utilization of the continuation-ratio logit model combined with the flexibility of generalized additive modelling enabled the characterization of multiple age patterns, and various linear and non-linear relationships within the data. This combination showed a valuable and insightful approach for comprehensively analysing age distributions and understanding the factors influencing the distribution of different age categories within the several cohorts available in our data. While some experiments were conducted by introducing interactions between time and spatial predictor variables, this posed computational challenges because of the dataset complexity and the model already having 50 estimated parameters. The potential impact of these interactions on model predictions proved difficult, and despite our best efforts, it remains unclear whether additional data points or a rescaling of the data to a different time and spatial frame may be necessary. However, it is important to note that further analysis should be undertaken to test the potential impact of interactions on the model predictions that can potentially improve the overall model predictions. While our modelling did not explicitly account for age-0 abundances, we also consider the potential impact of variations in year-class strength on our predictions of subsequent age probabilities. Exploratory modelling and the overall similar patterns of age distributions suggested consistent patterns of distribution in the analysed period and no evidence of a significant year/cohort effect in our estimated conditional age probabilities. Azevedo and Silva [24] explored the vessel effect using GLM/GAM mixed random effect models on the proportion of adults. However, the introduction of a random vessel effect did not significantly improve the models. Considering both the lack of improvement and the computational complexity associated with incorporating this effect into CRL models when there are more than a couple of terms in the vector of random effects [45], we opted not to further pursue this aspect.

This study acknowledges potential weaknesses, such as using horse mackerel effort and catch data from the bottom-trawl fishery and under-sampling of certain regions and depths. In fact, model fitting excluded horse mackerel data from the southern area and from depth strata deeper than 500 m depth. It is noted, however, that horse mackerel catches from the Portuguese trawl fishery come mainly from the western coast (NW and SW areas) at depths <500 m, and also that the analysis of the fishery strategy revealed a rather stable catchability over the analysed period. Moreover, while horse mackerel is commonly categorized as a pelagic fish, its behaviour on the Portuguese and Spanish coasts deviates from typical pelagic species. In these regions, it is primarily captured using bottom-trawl gear and assessed through scientific bottom-trawl surveys, such as the ICES IBTS surveys. Research on the species distribution implies a demersal behaviour, especially

during daylight hours, as observed in studies by Murta (2008) [8]. Dietary analyses further reinforce the species' close association with the seabed as horse mackerel primarily consumes organisms likely obtained near the seabed, suggesting a stronger association with the sea floor compared to typical pelagic species [46]. Therefore, utilizing bottom-trawl surveys for sampling horse mackerel was considered appropriate, without introducing significant bias to the data used in this study. The trawling fleet is the most important fishery for this species; nevertheless, catches are also obtained from other fleets operating in coastal areas (purse seine) and deeper areas (gillnets) [26], as fishing intensity can be distributed across a wide range of age groups. Enhancing the abundance index for younger age groups, particularly age-0, could be achieved by incorporating and applying the same framework used in this study to the purse seine fleet. Similarly, refining abundance indices for older age groups could be accomplished through the analysis of vessels employing gillnets that catch older and larger horse mackerel and enlarging the spatial scale to the northernmost and southernmost limits of the stock distribution. Still, the analysis of fine-scale commercial data analysis has the potential to mitigate the shortcomings of scientific survey data, where a notable weakness lies in the uneven distribution of sampling across time, resulting in potential under-sampling of specific time periods.

Spatio-temporal analyses play a crucial role in accounting for variability in commercial sampling over space and time, since ensuring that the indices of abundance consider the spatial and temporal distribution of effort is essential for accurate abundance and composition estimates [33]. Using the whole spatial resolution of the data in the analysis can be problematic due to a small number of samples and missing data in particular areas, as is commonly observed in commercial data. In addition to spatial variability, we also considered the temporal distribution of fishing effort. This study also revealed a strong weekly cycle of fishing effort related to auction market economic cycles. This is particularly important as fishing activity may fluctuate seasonally or over different time periods, leading to variations in abundance and age composition data. Mitigating this variability successfully identified some seasonal patterns and addressed a potential bias in using commercial effort data. This emphasizes the importance of considering temporal dynamics and filtering data with specific timeframes. By accounting for these temporal dynamics, we can better understand the changes in abundance and composition over time. To mitigate this issue, a straightforward simple average smoothing method has been successfully implemented to reduce bias and deliver accurate information for the specific time (Julian week) and spatial scales (geographic areas) examined in this study. By incorporating these considerations, we provide a comprehensive framework for addressing the variability in effort, leading to more robust and accurate estimates of age abundance composition data for stock assessment purposes.

This study reveals a certain stability in horse mackerel age composition distribution, indicating a relatively constant population during the period 2010–2020, especially along the western Portuguese coast. Despite small variations, the analysis of catch-at-age per unit effort suggests seasonal patterns in the age composition, attributed to horse mackerel behaviour and availability to fishing gears that could be related to the potential impact of spawning and feeding migrations on fishing yields. Geographical differences in the presence of juveniles and adults are observed, suggesting variations in the migratory behaviour across different areas. Ultimately, the findings suggest that horse mackerel may have a more continuous distribution throughout the year in the western area, with some regional variations and possible connections to migratory patterns to adjacent areas. Although the southern limit of the stock might be inadequately sampled, the observed complex age distribution patterns in the southern area may signal a mixed population structure, challenging the notion of a uniformly continuous stock in this region, as suggested by earlier studies. The southern limit of the southern horse mackerel stock is not as evident despite a previous study indicating that the populations off the north of Africa and the Iberian Peninsula are not part of a continuous stock [7].

The migratory patterns of horse mackerel stock in the region presently designated as the southern stock appear to be more intricate as outlined in the data analysed in this study. Achieving a more detailed analysis of these behaviours is currently challenging, and this could be improved with increased temporal coverage and the incorporation of additional fleet information to enhance spatial coverage. The potential resolution of this limitation in the future involves integrating combined information from bottom-trawl, purse seine, and polyvalent fleets, offering a more comprehensive understanding of horse mackerel migrations along the Atlantic coast of the Iberian Peninsula. While tagging could also yield valuable insights into the movements of horse mackerel, it is worth noting that previous attempts have highlighted technical challenges in achieving an adequate survival rate for tagged individuals [7,8]. Our analysis also reinforces horse mackerel ontogenic deepening with increasing size and age, as suggested by Murta (2008) [8] using a time series of horse mackerel abundance from survey data and by Azevedo and Silva (2020) [24] by modelling the variation in the daily proportion of adult fish of the Portuguese bottom-trawl catches in the single year of 2017. The latter study revealed interesting patterns in the distribution of adult and juvenile horse mackerel where the proportion of adult fish was found to increase from shallow waters up to a depth of approximately 220 m, and then slightly decrease thereafter. Additionally, the proportion of adult fish increased from January to June, coinciding with the main spawning season of horse mackerel off the Portuguese coast.

These previous studies mainly differentiated juvenile and adult distributions using a length threshold or an estimated maturity ogive. A more detailed analysis in this study revealed similar patterns but with a finer examination of age groups. From the analysis of the predicted probabilities of the CR model, age-1 (entirely juvenile) and age-2 (64% juvenile) exhibit seasonal and depth patterns, showing increased abundance in the autumn and winter seasons and a decline in abundance with depth until 150 m. Age-3 lacks a clear pattern throughout the year and by depth, likely due to a mix of behaviours from both adult and juvenile (18%) individuals in this age category. In a previous study, which suggested a uniform distribution of juveniles throughout the seasons, the differentiation between juvenile and adult distribution was based on a specific length threshold [12]. Nevertheless, through a more detailed analysis of age composition in our study, it becomes evident that these patterns may involve contributions from different age classes with varying expected probabilities. Adult populations from age-4, which are almost fully mature, follow a depth pattern similar to previous adult observations but lack a discernible seasonal trend. For age-5 and older, the pattern aligns with findings from the prior study, indicating increased abundance with depth, a somewhat decreasing trend with depth in older ages, and increased abundance from the beginning of the year until early summer, also aligning with the spawning season of horse mackerel. The stratified nature of the population resulting from ontogenic migrations, with younger individuals predominantly occupying shallower strata and gradually shifting to deeper strata, can be shown in age specific distributions.

This study revealed interesting insights into horse mackerel distribution range and age specific distribution from the Northwest (NW) area to the Southwest (SW) area. The findings indicate a pattern of age specific distribution, with the NW area characterized by a concentration of younger individuals, with a gradual shift in abundance observed towards the SW region from age-3 to age-6. Older individuals seem to balance in both regions, and this could indicate a return of older individuals to the NW area in the spawning season. These distributional patterns align with previous studies [8,9] based on scientific survey data in the studied area. There are indications that these older individuals are also more available in the northernmost distribution of the stock based on the Spanish IBTS survey in Galician waters and the catch profile of trawlers operating in the region [26]. The consistent trends and patterns observed across various year classes suggest the presence of migratory behaviours in horse mackerel.

While specific age-related distribution patterns exist, there is also a noticeable and consistent presence of all age groups in various areas and depths. This suggests a widespread

age distribution and stable abundance and consistency in the fishing grounds. This stability points to a relatively constant population during the period 2010–2020. Villamor et al. (1997) [47] similarly noted that the horse mackerel fishery in the southernmost part of the Bay of Biscay occurs along the entire continental shelf throughout the year, with only slight spatial-seasonal variations in landings.

Horse mackerel, characterized by high genetic variability, possess a broad foundation for adapting to various conditions and stressors. Additionally, the species' notable vagility and gene flow indicates potential population connectivity and the ability to move about freely and migrate [7] which could also add to the species' resilience in response to environmental and fishing pressures. Additionally, there is evidence of sustained reproductive capacity across the observed Spawning Stock Biomass (SSB) levels [26], with the species employing a multiple batch spawning strategy [13,48]. This strategy, distributing the risk to their offspring on a temporal and spatial scale as a more successful risk-spreading strategy, proves to be effective, especially in the context of diverse gear selection patterns [49]. In fact, this is the particular case of the southern horse mackerel stock which is explored by different gears with complementary size selection patterns [26]. These aspects collectively indicate the resilience of horse mackerel as a species, reflecting its ability to maintain recruitment, a widespread distribution, and a rather stable abundance which could reflect in the absence of clearly defined age distribution patterns even in the presence of detailed fine-scale resolution data.

There are several studies that collectively suggest that environmental factors can play a role in shaping the population dynamics and distribution of horse mackerel populations. The influence of several environmental factors, such as sea surface temperature (SST), wind components, and oceanic transport indices can shape the distribution and spawning areas of this fish species [7,50,51]. Growth and maturation can also be influenced by the photoperiod, and winter upwelling along the Portuguese coast can negatively affect recruitment [52]. Our findings emphasize the need for further investigations to disentangle ecological factors influencing horse mackerel age distributions patterns and their implications for fisheries management. Furthermore, employing statistical analyses that incorporate multiple categories in the response variable, as in our analysis, is meaningful to identify the key aspects driving the dynamics of horse mackerel populations. The framework used in this study could enhance the importance of understanding the relationship between environmental variables and horse mackerel dynamics in the North Atlantic and Iberian waters, emphasizing the need to consider factors such as temperature, wind patterns, and oceanic circulation in future studies. The impact of these environmental variables on the mortality rates and distribution of fish species could shed light on the complex interplay between climatic and oceanic conditions and the distribution and survival of life stages, notably the early life stage, of this fish species.

Although our study focused on the age distribution patterns of horse mackerel, the method used for estimating time series of fishing-effort distributions from VMS data could have several potential applications in fisheries and environmental assessment and management. For example, it could be used to identify areas of high fishing effort and to assess the impact of fishing on the environment. It could also be used to evaluate the effectiveness of marine protected areas and to inform the design of spatial management measures. Additionally, the method could be used to compare fishing effort across different gears or over time, and to support the development of ecosystem-based fisheries management. There are implications of these findings for the management and conservation of horse mackerel populations in the Atlantic. The migrations of horse mackerel should be considered when designing management strategies, as well as the differences in recruitment patterns between different areas and depths. This study provides valuable information for understanding the ecology and dynamics of horse mackerel populations and could be used to inform future research on the species, contribute to the development of improved methods for defining stock units, and ultimately contribute to a more informed fisheries management.

## 5. Conclusions

The methodology employed in this study, particularly in estimating spatial time series of fishing effort and catch-at-age distributions, holds broader applications for fisheries assessment and management. The utilization of the continuation-ratio logit model, designed for ordinal categories, proved to be a suitable analytical tool, overcoming structural issues encountered by other models when analysing large fine-scale commercial data with multiple response variable categories. Despite potential weaknesses in data collection, such as under-sampling and exclusion of certain regions, this study provides valuable insights into the complex age distribution patterns of horse mackerel in the Atlantic waters off the Iberian Peninsula. The findings suggest a stable age composition over the period 2010–2020, with some indications of migratory behaviours with regional variations likely connected to spawning and feeding migrations. There is also evidence of a stable ubiquitous presence of the population across different areas and depths, indicating a broad foundation of this species for adapting to various conditions, probably related to important life history traits such as a notable vagility, spawning strategies, and adaptable feeding behaviour. The observed stability and consistent landings over an extended period suggest that future changes in fishing strategies for this stock are not anticipated. This finding is a significant result from our study that could be enhanced significantly by incorporating other Northeast Atlantic areas where this species is explored. This broader inclusion would contribute to multidisciplinary studies on stock identification and ecological factors influencing age distributions. In conclusion, this research contributes to a more informed understanding of horse mackerel ecology and dynamics, providing valuable information for the sustainable management of these populations in the Northeast Atlantic.

**Supplementary Materials:** The following supporting information can be downloaded at https://www.mdpi.com/article/10.3390/fishes9030093/s1, Supplementary Material_Data: Figure S1: Distribution of horse mackerel landings in percentage from 2010–2020 by fishing gear, Figure S2: Distribution of horse mackerel landings in percentage from 2010–2020 by area, Figure S3: Distribution of horse mackerel landings from 2010–2020 by commercial size category, Figure S4: Mean length distribution by commercial size category from 2010–2020, Figure S5: Fishing effort by area from 2010–2020, Figure S6: Proportion of fishing effort from 2010–2020 by area and depth, Figure S7: Spatial distribution of fishing effort from 2010–2020, Figure S8: Mean catch in number of individuals by age-0 to age11+ from 2010–2020, for exploratory data analysis on catch and effort; Supplementary Material_Model: Table S1: Summary of CR models tested with VGLM and VGAM functions, Table S2: Detailed results of the VGAM function for the selected model, Figure S9: Partial effect for categorical variable area, Figure S10: Partial effect for variable Julian week, Figure S11: Partial effect for variable depth, for VGAM outputs of the Continuation Ratio Logit model results.

**Author Contributions:** Conceptualization, H.M., M.A. and C.S.; data curation, M.A. and C.S.; formal analysis, H.M.; writing—original draft preparation, H.M. All authors have read and agreed to the published version of the manuscript.

**Funding:** This research received no external funding.

**Institutional Review Board Statement:** The species *Trachurus trachurus* is not listed as threatened, endangered, or protected on either the IUCN Red List or any national/regional lists or schedules of protected fauna, and the specimens reported in this paper were not collected in any protected area. Specimens were collected under the Portuguese National Biological Sampling Program in accordance with the European Union Data Collection Framework Regulation (EU) 2017/1004 for the collection, management, and use of data in the fisheries sector and support for scientific advice.

**Informed Consent Statement:** Not applicable.

**Data Availability Statement:** The data presented in this study are available on request from the corresponding author. The data underlying this article cannot be shared publicly due to VMS data provider confidentiality conditions. The data will be shared on reasonable request to the corresponding author.

**Acknowledgments:** We thank all the collaborators involved in the collection of biological data at the auction markets. We also like to thank the Portuguese Directorate of Fisheries (DGRM) for providing detailed landings and VMS data. This study was supported by the Portuguese National Biological Sampling Programme from the EU Data Collection Framework (PNAB/DCF).

**Conflicts of Interest:** The authors declare no conflicts of interest.

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
