# Peer review of "Southern Horse Mackerel (Trachurus trachurus) Spatio-Temporal Distribution Patterns Based on Fine-Scale Resolution Data"

_fishes, doi:10.3390/fishes9030093_

Round 1
Reviewer 1 Report
Comments and Suggestions for Authors
The paper describes a method for investigating the distribution of southern horse mackerel in space, time and by age class.
It is an interesting paper and it falls within the scope of the journal; I think the paper will be of significant interest to the readers of Fishes and I would like to see this work published. However, I have some concerns about the data processing and modelling. Possibly, these can be addressed by clarifying the text but it is also possible that there are substantial issues with the approach. Therefore, I cannot recommend to accept the paper with only minor revisions; I think another round of reviews is necessary.
The structure of the paper is logical but some detail is missing (see detailed comments). The references appear to be relevant and up-to-date. The number of figures and tables is appropriate although some of the figures could be improved somewhat or even omitted (see detailed comments).
Detailed comments
The first para on page 3 introduces the bottom trawl fishery but does not state that this is a targeted fishery and the dominant horse mackerel fishery. Naively I assumed that a pelagic fishery would be the dominant fishery for this species. This is discussed later but initially It was unclear to me why the paper focused on the demersal fishery.
Figure 1. It is not clear exactly what the ‘Portuguese’ area is from this figure. In the north it appears to be bounded by the northern boundary of Portugal but in the south it is not clear whether the area of Cadiz is considered part of the study area until we see Figure 3. Also, the figure might be clearer if the land was coloured in (e.g. grey like Fig 3) to distinguish it from the sea area. The labels on the depth contours are also too small to read, either make them larger or describe in the figure header that the 100, 200 and 300m contours are shown.
3rd para on page 4. It might be useful to say a bit more about the length data without the reader having to refer to the Azevdo papers. E.g. are the length composition data in 1cm size classes? How good is the coverage of the sampling scheme in time and space? Are there any trends or patterns in the data? Some of this is addressed in the supplementary material so a couple of sentences should be enough.
3rd para on page 4. Is it necessary to remove trips that fished in more than one area? Is there potentially a bigger bias resulting from removing these trips than from assuming that the catch on a trip in multiple areas had the same size distribution?
3rd para on page 4. The depth grid is provided at a 1-minute spatial resolution but presumably aggregated to 0.05 degrees (table 1)? This is not clear from the text.
3rd para on page 4. One of the more serious concerns I have is with the ALK approach. Assigning length distributions to trips based on commercial size classes seems justifiable. However, it is unclear to me whether the ALKs are for all years combined or specific for each year (and semester). If all years are combined, then the estimated number of fish in strong cohorts will be under-estimated and weak cohorts will be over-estimated. This is because the relative abundance of cohorts affects the proportions-at-age in each length class (in the ALK). This is particularly a problem for age classes where there is a lot of overlap in length, say from age 5 onwards for horse mackerel. Perhaps a more serious problem is that the ALK does not account for spatial differences, which clearly do exist. These two issues combined result in “age” classes that have lost some of their spatial and temporal contrast (it has been smoothed out by assuming a generic ALK). Instead what you end up with are essentially a set of size classes of fish that – on average – are around the size of a fish of age 1, 2, 3 etc. (what I mean is that when you talk about, say, age 10 fish, these are fish that are around the size of an average age-10 fish; there is no real way of knowing if they may be large age 9 fish if that cohort is much more abundant than the fish that are actually age 10 at the time). This approach is still useful and, as the result demonstrate, there is still enough contrast left to discern spatial and temporal patterns, however I think that this caveat should be acknowledged (I don’t think it can be fixed without a very detailed sampling programme). Perhaps it is more constructive to turn this around and use the results to justify why age sampling should be spatially stratified.
2nd para on page 6. The explanation of the difference between ACF and PACF is not clear. Also, there are many reasons why one would expect autocorrelation from one day to the next: i.e. many trips are presumably longer than one day; weather and other conditions are auto-correlated etc. It seems like the main purpose of this analysis is to show a weekly pattern. I think that this section can be reduced considerably. Even a table or boxplot of effort by weekday would show the weekly pattern.
3rd para on page 6. The variogram shows a cyclical pattern but there is no real underlying structure that can be accurately described as cyclical. There are a bunch of fishing grounds that happen to be spaced out around 30-60nm. Also the width of the continental shelf is around 30nm which in itself may result in autocorrelation at the eastern and western end (both with low effort with high effort in the middle). I would also question the use of an omnidirectional variogram when there is likely to be anisotropy. I would remove the reference to the variogram.
3rd and 4th para on page 6. Both paragraphs provide a (slightly different) reason for omitting the southern area. I would combine this into a single paragraph.
1st para on page 8. Abundance indices are calculated but it not explicitly stated what the units of effort are here (presumably KW*hours like in the exploratory analysis). More detail is needed here, how did the authors deal with grid cells with no or insufficient data? Also, vessel effects are usually a significant factor in CPUE indices; perhaps some of the differences between vessels is accounted for in the spatial aspect but it would be good to have a bit of text on this. Including a vessel effect in the model would be a good way to address potential bias here. Finally, it would be good to be explicit about the purpose of calculating these CPUE indices. There are some hints in the paper that this will be relevant for stock assessment but that is not the purpose in this analysis.
3rd para on page 8. What exactly is the response variable? Not the age group, is it the proportion at age in each grid cell? Again: what about missing grid cells? More importantly: what about spatial autocorrelation between grid cells? Is each grid cell treated as an independent observation? That would presumably be a considerable violation of the model assumptions. How can the authors justify this?
3rd para on page 8. “Y” is not defined.
Last para on page 8. How is disproportionate quantified?
3rd pare on page 9. What are “sampling commercial vessels”? Also: Fig 4 does not show age selectivity. CPUE at age varies with the abundance at age while selectivity is independent of abundance.
Figure 5. The medium and light grey are quite similar (on my screen) it would be better with more distinct shades (black, grey white).
Equation 2. No year effect is included. The ICES assessment for this stock shows that there is a fair level of variability in recruitment from year-to-year (it varies by a factor of around 2). So the authors need to provide a good argument for not including a year effect. Also, why is the depth effect the same for all age classes when the preliminary analysis showed an interaction between depth and age?
First para on page 16. Does the fishery occur mainly during daylight hours?
First para on page 16. “Enhancing the abundance index” – is the ultimate purpose to create abundance indices for stock assessment? If not what are you trying to enhance.
First para on page 19. “The findings suggest a stable age composition distribution over the period 2010-2020” – this contradicts the estimated annual recruitment from the ICES stock assessment, which indicates considerable differences in cohort strength during that period. The current method does not really allow to identify changes between years as the ALK appears to be combined over all years and the model does not include a year-effect.
Comments on the Quality of English LanguageThe quality of the writing is generally very good and clear; some minor language edits are needed though.
Reviewer 2 Report
Comments and Suggestions for Authors
See my report.

Reviewer 3 Report
Comments and Suggestions for Authors
The manuscript of Mendes et al. describes a very interesting methodology to investigate and monitor stock dynamics and fishing efforts. Authors have analyzed long-term data regarding landings and fishing efforts of a very ecologically and commercially valuable species.
I suggest some minor revisions to make the paper more clear and readable.
Authors should avoid in the MeM chapter any results presentation or introductory speech. Concerning results, they should avoid any results comments.
A pdf file of the paper with some suggestions is provided attached to this comment.

English needs only some minor editing on language
Reviewer 4 Report
Comments and Suggestions for Authors
The study provides crucial insights into the challenges associated with commercial fishery data collection and patterns in the abundance of specific age groups of Trachurus trachurus. Comments:
1. Necessary to Title Refinement: Consider revising the title to be more detailed and informative.
2. Although a lot of work has been done. The trends and prospects for fishing in the future are unclear. Since there is no clear statistical comparative treatment. Trends need to be described more clearly. The age dynamics are unclear.
3. Statistical analysis:
· There is too much scatter in the data. Probably the data collections were not equal in different years. It is necessary to bring the samples to the same scale so that they can be compared. Remove unnecessary data and artifacts (fig. 4).
· Further analysis is needed to scrutinize the potential impact of interactions on model predictions for potential improvements in overall fit.
5. Provide more specific information and numerical data in the conclusions section to enhance clarity. Explicitly detail the observed age distribution patterns, making it clear whether they align with expectations or if there are deviations. Specify numerical values, trends, or percentages to support the conclusions drawn.
Comments on the Quality of English Language
A slight correction of the English language is required.
Round 2
Reviewer 1 Report
Comments and Suggestions for Authors
The authors have clarified the issues raised in the first round of reviews. My concerns about the methods have been addressed by these clarifications in the text. I have a few minor comments:
Line 62 "Previous studies [12,13] ..." - I would put the citations here and then change the following sentence to something like "A study based on .... [12] analysed the..." and the same for the next study.
Line 105. "This study" might refer to Azevedo & Silva. Maybe better to say: "The current study"
Line 117. I would say: "The framework used here has..."
Line 395. This is not really selectivity; the catch numbers at age are determined by the selectivity (the catchability-at-age) as well as the population numbers (which decline with age due to natural and fishing mortality). I would avoid the word 'selectivity' and replace it by 'age profile'.
Lines 592-597 - these are repeated from lines 502-506
I found the discussion rather long; I did not raise this in the first review so I will not insist on a major re-write but it may help the accessibility of the paper to focus the discussion on the main points of interest only.
Comments on the Quality of English LanguageThere are a small number of grammatical errors that a spell-checker will pick up.
Reviewer 3 Report
Comments and Suggestions for Authors
The MS has been improved according to the reviewer's suggestion.
Comments on the Quality of English LanguageThe quality of English language is good.
Author Response
Dear Reviewer,
We revised several English spell-check suggestions in lines 54, 188, 196, 199, 456, 541, 628, 752, 774, and 706, primarily related to UK/US English variations (e.g., 'analyzed' for 'analysed,' 'programme' for 'program'...).
Additionally, we refined English sentences in lines 62, 105, 117, and 345, and eliminated a similar/repeated sentence at the end of the results section. We are grateful to the reviewer for the adjustments pointed out, which significantly contributed to clarifying key points in the study and improved the overall structure of the main text.
Best regards